# The Modulation of Arachidonic Acid Metabolism and Blood Pressure-Lowering Effect of Honokiol in Spontaneously Hypertensive Rats

**DOI:** 10.3390/molecules27113396

**Published:** 2022-05-25

**Authors:** Fawzy Elbarbry, Nicholas Moshirian

**Affiliations:** School of Pharmacy, Pacific University, Hillsboro, OR 97123, USA; mosh6645@pacificu.edu

**Keywords:** honokiol, hypertension, arachidonic acid, epoxide hydrolase, natural products

## Abstract

Background: Cardiovascular diseases have consistently been the leading cause of death in the United States over the last two decades, with 30% of the adult American population having hypertension. The metabolites of arachidonic acid (AA) in the kidney play an important role in blood pressure regulation. The present study investigates the antihypertensive effect of honokiol (HON), a naturally occurring polyphenol, and examines its correlation to the modulation of AA metabolism. Methods: Spontaneously hypertensive rats (SHR) were randomly divided into four groups. Treatment groups were administered HON intraperitoneally at concentrations of 5, 20, and 50 mg/kg. Blood pressure was monitored at seven-day intervals. After a total of 3 weeks of treatment, the rats were euthanized and the kidney tissues were collected to examine the activity of the two major enzymes involved in AA metabolism in the kidney, namely cytochrome P450 (CYP)4A and soluble epoxide hydrolase (sEH). Results: Rats treated with HON did not experience the rise in blood pressure observed in the untreated SHR. High-dose HON significantly reduced blood pressure and inhibited the activity and protein expression of the CYP4A enzyme in the rat kidney. The activity of the sEH enzyme in renal cytosol was significantly inhibited by medium and high doses of HON. Conclusion: Our data demonstrate the antihypertensive effect of HON and provide a novel mechanism for its underlying cardioprotective properties.

## 1. Introduction

Cardiovascular diseases (CVD) have been consistently ranked as the leading cause of death in the United States over the last two decades. Recent statistics indicate that approximately 40% of the entire US population has at least one type of CVD; of these, 80% have hypertension [1]. Of the adults with hypertension, only approximately 50% have their blood pressure controlled to <140/90 mm Hg and, therefore, are at higher risk of cardiovascular morbidity and mortality as well as an increased dependence on health-care resources [2]. Hypertension was listed as the leading or a contributing cause of death and resulted in a death rate of 19.9 per 100,000 people [2] in 2019, and the costs of treating hypertension in the US in that year were USD 51.2 billion in direct medical costs and USD 3.9 billion in indirect costs owing to lost productivity related to morbidity and mortality [1]. The American Heart Association (AHA) estimates that the total cost of treating hypertension in the US in 2030 will be USD 200–225 billion. In addition to those with hypertension, approximately 30% of American adults have elevated blood pressure and, therefore, have a greater risk of developing hypertension and total CVD events [3]. Thus, newer strategies for managing hypertension are greatly needed.

Although several classes of antihypertensive agents are available, adherence to pharmacological therapy for hypertension is low (the estimated range is 50–70%). A significant portion of patient treatment non-adherence may be attributed to adverse effects, cost of medication, inefficacy, or inconvenience [4]. Although the acquisition costs of some drug classes, e.g., diuretics, are low, when the high costs of office visits, lab tests, and follow-up for adverse effects are accounted for, there is no significant difference in medication costs across all drug classes. Non-adherence to antihypertensive medications results in poor control of hypertension, which is normally associated with higher costs due to increased cardiovascular morbidity and mortality. Newer and safer agents that can be used alone or in combination with pre-existing medications, thereby reducing the necessary therapeutic dose, would have a positive impact on the treatment and management of hypertension and pre-hypertension.

The AHA performed a literature review in 2013 of several studies involving alternative modalities of hypertension treatment and concluded that “it is reasonable for all individuals with blood pressure levels >120/80 to consider trials of alternative approaches as adjunct methods to help lower BP when clinically appropriate” [5]. Recent research from our lab has demonstrated that chronic administration of the polyphenolic compound quercetin to young spontaneously hypertensive rats (SHR) results in preventing the progressive rise in blood pressure in this animal model [6]. Since polyphenols are natural products and rich in the diet, they can provide a viable option for patients with elevated blood pressure. It is known that simple dietary constituents have the potential to impact human health and significantly reduce the cost in health care [7].

Several studies have indicated that arachidonic acid (AA) metabolites play a critical role in the regulation of renal vascular tone, tubuloglomerular feedback, and sodium transport [8]. Renal cytochrome P450 (CYP)-mediated metabolism of AA produces either hydroxyeicosatetraenoic acids (HETEs; particularly, 19- and 20-HETE) or epoxyeicosatetraenoic acids (EETs; Figure 1). In the renal and peripheral vasculature, 20-HETE is a potent vasoconstrictor and is involved in tubuloglomerular feedback and the autoregulation of renal blood flow and the glomerular filtration rate [9]. On the other hand, EETs are recognized as endothelium-derived hyperpolarizing factors (EDHF) with a multitude of biological activities, including renal vasodilation and antihypertensive effects [10]. However, EETs are rapidly metabolized by soluble epoxide hydrolase (sEH) to their corresponding diols, which lack the biological activities of their precursor EETs [11]. Therefore, targeted inhibition of CYP4A and/or sEH enzymes is a promising approach for the treatment of hypertension and the restoration of the dilation function of the endothelium. We have previously demonstrated that inhibition of CYP4A is associated with reduced formation of 20-HETE and reductions in mean arterial blood pressure [12]. Studies by our laboratory and others demonstrate that chronic inhibition of sEH for 1–6 weeks lowers blood pressure and ameliorates the organ damage associated with hypertension [13]. As most hypertensive patients are likely to receive several medications to treat other medical conditions associated with hypertension (e.g., kidney damage, diabetes, and heart failure), specific inhibition of sEH is an attractive target for lowering blood pressure with a minimal risk of drug–drug interactions since sEH is not primarily involved in drug metabolism.

Honokiol (2-(4-hydroxy-3-prop-2-enyl-phenyl)-4-prop-2-enyl-phenol; HON; Figure 1) is the major active polyphenol constituents of magnolia bark (*Magnolia officinalis*), which has been widely used in traditional Chinese medicine for the treatment of various diseases, including anxiety, stress, gastrointestinal disorders, infections, and asthma [13]. Human exposure to a wide range of Honokiol doses through either nutraceuticals or chewing gums indicated wide safety margins [14].

The current study was undertaken to examine the antihypertensive effect of short-term administration of HON in a hypertensive rat model and to investigate its effect on AA-metabolizing enzymes in the kidney.

## 2. Results

Daily intraperitoneal administration of HON for 3 weeks had no obvious adverse effect on the rats. As illustrated in Figure 2, there were no significant differences among the study groups in terms of age-related increase in body weight. Additionally, the average daily water intake was assessed for each cage and determined to be 19.5 ± 1.5 mL/100 g of body weight (*p* > 0.05).

### 2.1. Effect of Honokiol Treatment on Blood Pressure

During the three-week acclimation period, blood pressure was measured weekly. As demonstrated in Figure 3, there was no statistically significant difference in baseline blood pressure among the rats in all study groups prior to the initiation of HON administration. As expected, all untreated rats showed a progressive rise in blood pressure that is characteristic of the developmental phase of hypertension in SHR. For example, the control group, low-dose HON group, medium-dose HON group, and high-dose HON group exhibited 11.9%, 11.4%, 9%, and 8.8% increases in SBP, respectively, during the three-week acclimation period (Week −3 to Week 0; Figure 3A). To examine the effect of daily i.p. administration of HON on blood pressure in SHR rats, we measured the SBP, DBP, and MAP of all groups once weekly for 3 weeks (Week 1 to Week 3). The SBP increased by +5.9%, +7%, and +11.2% in the control group in Week 1, Week 2, and Week 3, respectively (Figure 3A). Animals treated with the low-dose HON (5 mg/kg) exhibited changes in SBP of +3%, 0%, and −3.6% in Weeks 1, 2, and 3, respectively (Figure 3A). Administration of a medium dose (20 mg/kg) of HON resulted in a change in SBP of +1.8%, −1.8%, and −6.2% in Weeks 1, 2, and 3, respectively (Figure 3A). Rats treated with HON at a high dose (50 mg/kg) demonstrated changes in their SBP of −4.8%, −10.2%, and −14.3% in Week 1, 2, and 3, respectively (Figure 3A). Compared to the control group at Week 3 of treatment, SBP was reduced by 15%, 19.5%, and 23.8% by low-, medium-, and high-dose HON, respectively. Despite the observed improvement in SBP in the treatment groups at Week 3 of treatment, a statistically significant difference was only observed in the medium- and high-dose groups compared to the control group.

Similar to the observed progressive rise in SBP, all rats exhibited a continual rise in their MAP during the three-week pretreatment period. Specifically, MAP increased by 12.7%, 12.8%, 13.7%, and 11.7% in the control group, low-dose HON group, medium-dose HON group, and high-dose HON group, respectively. The control rats continued to exhibit a rise in MAP by 4.8%, 10.5%, and 14.5% after Weeks 1, 2, and 3, respectively (Figure 3B). Treatment with HON did not reduce the MAP but significantly reduced the progressive rate of its rise compared to the control rats. The MAP increased by 4%, 5.6%, and 8.9% in the low-dose group; by 1.6%, 3.2%, and 6.4% in the medium-dose group; and by 1.6%, 1.6%, and 4% in the high-dose group after Week 1, Week 2, and Week 3 of treatment, respectively (Figure 3B). Compared to the control group at Week 3 of treatment, MAP was reduced by 5.6%, 7.1%, and 8.5% by the low-, medium-, and high-dose HON treatments, respectively. The changes in MAP after 3 weeks of treatment were statistically significantly different from the control group only in the high-dose treatment group.

Similarly, the untreated rats showed a gradual rise in DBP during the three-week acclimation period, with increases of 8.7%, 6.2%, 7.4%, and 7.6% in the control group, low-dose HON group, medium-dose HON group, and high-dose HON group, respectively. The DBP continued to rise in the control group and increased by 3.5%, 9%, and 13.8% at the end of treatment Weeks 1, 2, and 3, respectively (Figure 3C). Treatment of rats with a low dose of HON reduced the rate of the observed rise in DBP in the control group by approximately 30% after Week 1 and Week 2 of treatment and by 18% after Week 3 of treatment (Figure 3C). The DBP increased by 1.1%, 2.2%, and 4.5% in the medium-dose group, and by 2%, 4.6%, and 5.7% in the high-dose group after Week 1, Week 2, and Week 3 of treatment, respectively (Figure 3C). Compared to the control group at Week 3 of treatment, DBP was reduced by 4.1%, 8.1%, and 9.1% by the low-, medium-, and high-dose HON groups, respectively. The DBP at Week 3 of treatment was statistically significantly lower in both the medium- and high-dose HON groups compared to the control group.

### 2.2. Effect of HON Treatment on 20-HETE Formation

To examine the potential inhibitory effect of short-term daily i.p. administration of HON on the 20-HETE formation rate as a marker for CYP4A activity, 20-HETE was measured in rat renal microsomes after three weeks of exposure to HON at low, medium, and high concentrations using an LC method, as described in the Methods section. Figure 4 illustrates the trend of dose-dependent inhibition of 20-HETE formation. However, a statistically significant reduction in the 20-HETE formation rate was only achieved after the high (50 mg/kg) dose. The formation of 20-HETE was inhibited to 90%, 82%, and 69% of the control value by the low, medium and high doses of HON, respectively (Figure 4).

### 2.3. Effect of HON Treatment on CYP4A Protein Levels

To examine the effect of HON on the expression level of CYP4A protein, Western blotting of renal microsomes was performed using a polyclonal antibody against rat CYP4A1/2/3. As shown in Figure 5a, Western blots performed using antibodies to CYP4A revealed a single band. Additionally, HON-mediated inhibition of AA ω-hydroxylation was associated with a loss of CYP4A immunoreactive protein in a dose-dependent manner (Figure 5b). Treatment with medium (20 mg/kg) and high (50 mg/kg) doses of HON significantly reduced the level of CYP4A apoprotein to 65% and 52% of the control value, respectively. On the other hand, no significant inhibition was observed in the low-dose HON group (Figure 5).

### 2.4. Effect of HON Treatment on Soluble Epoxide Hydrolase (sEH) Activity

To investigate the potential inhibitory effect of HON on the activity of sEH, an enzyme that hydrolyzes the cardioprotective effect of eicosanoids to inactive diols, we measured the activity of this enzyme in kidney cytosolic fractions from control and HON-treated SHR. The results shown in Figure 6 indicate that the sEH activity was inhibited by 30% and 44% (*p* < 0.05) following administration of medium (20 mg/kg) and high (50 mg/kg) doses of HON, respectively. Administration of the low dose (5 mg/kg), on the other hand, did not result in significant inhibition of sEH activity.

### 2.5. Correlation between HON Dose, CYP4A Activity, and sEH Activity

Due to the absence of CYP4A or sEH knockout models for SHR, establishing a cause–effect relationship between the observed effect of HON on MAP, CYP4A activity, and sEH activity was not possible with the present study. Therefore, we investigated the correlation between these variables. When HON-dependent changes in MAP were compared to changes in the 20-HETE formation rate as a measure by CYP4A activity, a good correlation was observed (Pearson r = 0.78 and *p* = 0.028). While the 5 mg/kg, 20 mg/kg and 40 mg/kg HON doses reduced MAP by approximately 5, 8, and 10.5 mmHg, respectively, the 20-HETE formation rate was reduced by 9, 18, and 31%, respectively (Figure 7). Similarly, changes in MAP reasonably correlated (Pearson r = 0.85 and *p* = 0.038) with changes in sEH activity in a HON-dose-dependent manner. Namely, compared to the 5, 8, and 10.5 mmHg reductions in MAP, sEH activity was reduced by 12, 31, and 44% following the administration of 5 mg/kg, 20 mg/kg, and 40 mg/kg doses of HON, respectively (Figure 7).

## 3. Discussion

In the present study, daily i.p. administration of honokiol (HON) for 3 weeks reduced systolic blood pressure and resisted the progressive rise in diastolic blood pressure in SHR, a rat model of hypertension. These effects were associated with the modulation of the arachidonic acid (AA) metabolism in the rat kidney, as represented by reductions in the activities of both CYP4A and sEH. We previously demonstrated similar effects with the isothiocyanate, sulforaphane [12], and the polyphenol, quercetin [6].

Honokiol is a promising bioactive polyphenolic compound with wide range of therapeutic activities, such as neuroprotective, antidepressant, anti-tumorigenic, antithrombotic, antimicrobial, and analgesic properties [13]. Several studies have investigated the cardiovascular protective effect of HON and attributed its mechanism of action to its potent antioxidant effects [15,16]. If the antihypertensive effects of HON are due exclusively to its antioxidant properties, then other antioxidants should also show similar antihypertensive properties. However, the results of three randomized clinical trials testing the effects of the antioxidant vitamin E on hypertension proved unsatisfactory [17,18,19]. Similarly, a non-significant reduction in blood pressure was reported in vitamin D–treated patients [20]. Accordingly, it is conceivable that HON and other antioxidants reduce blood pressure by more than one mechanism.

This study is the first to investigate the effect of HON on the AA-metabolizing enzymes in the kidney as a potential mechanism for its antihypertensive property. The study investigated the effect of HON on renal CYP4A, the enzyme responsible for the formation of the vasoactive AA metabolite 20-HETE. This metabolite is known to have a strong vasoconstrictive effect on the renal afferent arteriole that, in turn, is likely to trigger the powerful renin–angiotensin–aldosterone (RAA) cascade, one of the primary targets of antihypertensive therapy [21]. Additionally, we studied the effect of HON on the activity of sEH, which degrades the EET metabolites of AA that are potent vasodilators and natriuretic agents and have been proposed as endothelium-derived hyperpolarizing factors [10].

Our study demonstrated that HON reduces or prevents the progressive rise in blood pressure in SHR. Similar findings have been reported but used high-dose treatments (200 and 400 mg/kg) for 7 weeks [16]. Remarkably, the observed blood pressure–lowering effect by smaller HON doses after a three-week period was similar to the results achieved with commonly used antihypertensive medications in the same rat model. For example, losartan (15 mg/kg/day for 2 months), telmisartan (5 mg for 8 weeks), amlodipine (3 mg/kg for 12 weeks), hydrochlorothiazide (10 mg/kg for 8 weeks), and lisinopril (10 mg/kg for 8 months) resulted in average MAP reductions of 13%, 15%, 17%, and 20%, respectively [22,23,24,25].

Additionally, our study demonstrates an association between lowering blood pressure and a reduction in the formation rate of 20-HETE in kidney microsomes (Figure 4 and Figure 7) as well as reduced protein expression of CYP4A (Figure 5). It is well established that CYP4A is the principal enzyme responsible for the formation of 20-HETE, and the latter induces hypertension through its potent vasoconstriction in the renal and peripheral vasculature and by promoting sodium retention [8]. Substantial evidence has now accumulated signifying that the rapid rise in mean blood pressure in the SHR strain is, at least in part, attributed to the elevated production of 20-HETE in the kidney and mesenteric arteries [26]. This study is the first to illustrate the effect of HON on rat kidney CYP4A and 20-HETE formation. However, the presented findings support the hypothesis that inhibitors of CYP4A and 20-HETE synthesis reduce blood pressure and renovascular tone in SHR [10]. Using specific antibodies against CYP4A1/2/3, the immunoblotting data (Figure 5) indicate that HON lowers the expression of CYP4A isoforms in a manner similar to that involved in reducing MAP and 20-HETE formation.

Soluble epoxide hydrolase (sEH) is a key enzyme involved in metabolizing endogenously derived fatty acid epoxides and is considered an important therapeutic target in a wide range of human cardiovascular diseases. The combined inhibition of sEH and enhancement of the activity of the vasodilator EETs has been proposed as a promising approach for the prevention and treatment of hypertension and the prevention of organ damage. The present study indicates for the first time that both medium- and high-dose HON inhibit the activity of sEH in the kidney of SHR, and this inhibition was parallel to the observed reduction in blood pressure (Figure 6 and Figure 7). Since 2000, several large molecular weight ureas, carbamates, and amide inhibitors have been developed based on the x-ray structures of murine and human sEH enzymes [21]. Although these sEH inhibitors demonstrated potent antihypertensive effects, their poor physicochemical and unfavorable pharmacokinetic properties hindered their clinical use. The current study shows that naturally occurring phytochemicals such as HON can cause significant sEH inhibition and antihypertensive effects and can be presented as a cost-effective, stand-alone or complementary approach to the treatment of hypertension and the prevention of organ damage induced by uncontrolled high blood pressure. Our study, on the other hand, has a few limitations. The effects of honokiol in normotensive rats were not investigated. Therefore, it is unknown from our study whether honokiol in normal rats has a similar effect which could be translated to normal human individuals.

## 4. Materials and Methods

### 4.1. Materials

Honokiol (HON) was purchased from Sigma-Aldrich (St. Louis, MO, USA). Arachidonic acid, 20-HETE, and Epoxy Fluor 7 were purchased from Cayman Chemical Company (Ann Arbor, MI, USA). All reagents used for microsomal preparation, determination of protein content, and enzyme assays were purchased from Sigma-Aldrich (St. Louis, MO, USA). CYP4A1/2/3 polyclonal antibody was purchased from LifeSpan Biosciences, Inc. (Seattle, WA, USA). Acetonitrile and methanol were high pressure liquid chromatography (HPLC)–grade and obtained from Fisher Scientific (Pittsburg, PA, USA). All other chemicals used were analytical grade from VWR (Radnor, PA, USA).

### 4.2. Animals

Male 8-week-old spontaneously hypertensive rats (SHR) were obtained from Harlan Laboratories (Madison, WI, USA). All animals were maintained under controlled housing conditions of light (6 a.m.–6 p.m.) and temperature (22 °C) and received standard laboratory chow and water ad libitum. All rats were allowed at least 2 weeks to become acclimated to the housing conditions and ensure steady and reliable blood pressure readings before use in experiments. All procedures were approved by the Institutional Animal Care and Use Committee of Pacific University and Oregon Health & Science University.

### 4.3. Preparation of Honokiol Solutions

Stock solutions of 10, 40, and 100 mg/mL were prepared fresh daily in a vehicle of PEG400, ethanol, and saline at the ratio of 2:1:4 (*v*/*v*/*v*). Our pilot study indicated that these solutions had a pH of 7–7.5 and were stable for at least 3 days (data not shown). Honokiol solutions were filtered through a 0.2 µm membrane filter to ensure sterility before administration by intraperitoneal (i.p.) injection.

### 4.4. Honokiol (HON) Treatment

Following acclimation in the laboratory, animals were randomly divided into four groups of 8 animals each. Group one served as the control group and receive a vehicle of PEG400, ethanol, and saline at the ratio of 2:1:4 (*v*/*v*/*v*). Groups 2, 3, and 4 were treated with HON in vehicle at concentrations of 5, 20, and, 50 mg/kg, respectively. All rats received the vehicle or Honokiol by i.p. route once daily for 3 weeks. Animal body weights were measured daily and the HON dose (in mL) were calculated accordingly. The selection of HON doses was based on preliminary studies and previous studies [15].

### 4.5. Mean Arterial Pressure (MAP) Measurements

Systolic (SBP), diastolic (DBP), and mean arterial (MAP) blood pressure were measured in conscious, pre-warmed, restrained rats, as previously described [6,12]. For each SHR rat, blood pressure was measured once every week (baseline, Week 1, Week 2, and Week 3) using a non-invasive method based on determining the tail blood volume with a volume-pressure recording sensor and an occlusion tail-cuff (CODA system; Kent Scientific, Torrington, CT, USA). Our previous studies indicate that blood pressure values obtained with the non-invasive tail-cuff method are not significantly different from the values obtained in the same animals by direct measurements in the femoral artery. Blood pressure was measured by multiple readings in individual rats until an average of 15 stable measurements was obtained. Results are shown as the average of blood pressure values obtained from individual rats.

### 4.6. Tissue Collection and Preparation of Kidney Microsomes

At the end of the study, rats were anesthetized with isofluorane, the abdominal cavities were opened, and the kidneys were rapidly removed and rinsed with ice-cold saline. Kidney tissues were then flash-frozen in liquid nitrogen and stored at −80 °C until use. Kidney microsomes were prepared as described previously [6]. Microsomal protein concentrations were determined in triplicate using bovine serum albumin as a calibration standard, as described before [27]. Absorbance was measured at 750 nm on a Synergy2^®^ micro-plate reader using Gen5 Software (BioTek, Winooski, VT, USA).

### 4.7. Microsomal Arachidonic Acid (AA) Hydroxylation and Analysis of 20-HETE Formation

Oxidation of AA to its metabolite 20-HETE was determined in reaction mixtures (500 µL) containing 100 mM phosphate buffer (pH 7.4), 40 µM AA, 2 mM MgCl2, 1 mM NADPH, and rat kidney microsomes equivalent to 0.4 mg protein. Reactions were initiated with NADPH and were terminated after 15 min at 37 °C with 1.0 M HCl. The 20-HETE formation rate was measured as described previously, with few modifications [28]. Briefly, the reaction mixtures were extracted with ethyl acetate and the organic extracts were evaporated with nitrogen gas and the residues were reconstituted in the HPLC mobile phase. AA and 20-HETE were resolved on an Agilent Eclipse Plus C18 column (4.6 mm × 250 mm; Agilent Technologies, Santa Clara, CA, USA) with UV detection at 200 nm. Initial mobile phase composition was 45% acetonitrile in water with 0.1% acetic acid. Linear gradient (0.5%/min) was utilized over 30 min, followed by a sudden increase to 20%/minute to 100% acetonitrile. Flow rate was maintained at 0.3 mL/minute and column temperature at 40 °C.

### 4.8. Immunoquantification of CYP4A in Rat Kidney Microsomes

Rat kidney microsomes (10 µg per lane) were incubated at 100 °C for 5 min in Laemmli sample buffer (BioRad, Hercules, CA, USA) and electrophoresed 1 h constant voltage (150 V) through precast polyacrylamide gels (BioRad). Microsomal proteins were then transferred electrophoretically (30 min at 20 V) to PVFD membrane and incubated with CYP4A1/CYP4A2/CYP4A3 rabbit anti-rat polyclonal antibody at a 1:1000 dilution (Lifespan Biosciences) overnight at 4 °C. After washes with PBST, the secondary antibody (goat anti-rabbit conjugated to horseradish peroxidase, Thermo Scientific, Waltham, MA, USA) was added at a 1:5000 dilution for 1 h at room temperature. The immunoreactive proteins were detected via enhanced chemiluminescence and X-ray film imaging, and the resultant signals were analyzed by densitometry (Bio-Rad). The intensity of the band was normalized to GADPH signal, which was used as loading control.

### 4.9. Epoxide Hydrolase Activity (sEH)

Metabolism of Epoxy Fluor 7 to a fluorescent metabolite was utilized to determine the sEH activity in kidney microsomes, as described previously [6]. Briefly, reactions were carried out in mixtures (200 µL) containing 25 mM Bis Tris-HCl, 1 mg/mL BSA, Epoxy Fluor 7, and rat kidney microsomes (equivalent to 10 µg protein). The resulting solution was incubated at 37 °C in a black 96-well flat-bottom plate. The fluorescence of the Epoxy Fluor 7 metabolite was determined using an excitation wavelength of 330 nm and emission wavelength of 465 nm on a Synergy2^®^ microplate reader using Gen5 Software (BioTek)

### 4.10. Data Analysis

Data are reported as mean ± SD. Differences in 20-HETE formation rate and CYP4A protein expression were assessed by one-way analysis of variance (ANOVA) with Tukey’s post hoc test for pair-wise multiple comparisons. Differences in blood pressure among groups were analyzed using repeated measures ANOVA. Correlation analysis was also performed to investigate whether a change in blood pressure corresponded to a change in CYP4A or sEH activities. Statistical analysis was conducted using GraphPad Prism 5.0 (GraphPad Software Inc., San Diego, CA, USA). Tukey’s post hoc test was used for pair-wise comparisons. *p* < 0.05 was considered as statistically significant.

## 5. Conclusions

In conclusion, our findings have important clinical findings. The presented data indicate that dietary doses of honokiol have the potential to reduce and prevent the progressive rise in blood pressure in SHR. Additionally, these effects were associated with modulation of AA metabolism through inhibiting both the formation of 20-HETE and the activity of sEH. These results contribute novel knowledge about the antihypertensive properties of HON.

## Figures and Tables

**Figure 1 molecules-27-03396-f001:**
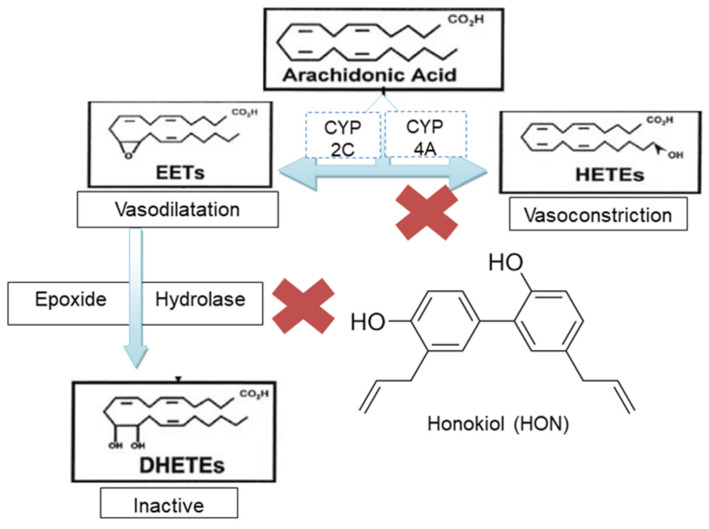
Metabolism of arachidonic acid by cytochrome P450 (CYP) enzymes and soluble epoxide hydrolase (sEH). The sign “x” indicates the proposed sites for the antihypertensive effect of HON.

**Figure 2 molecules-27-03396-f002:**
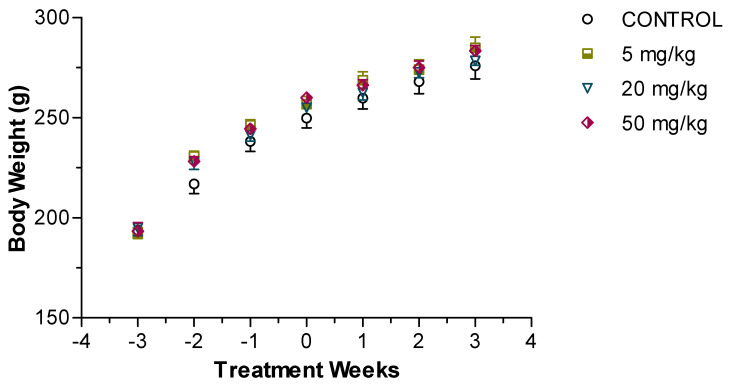
Average weekly body weight of control and HON-treated spontaneously hypertensive male rats. Data are presented as mean ± SEM, with each data point representing *n* = 8. No statistically significant differences were detected between control and treated rats.

**Figure 3 molecules-27-03396-f003:**
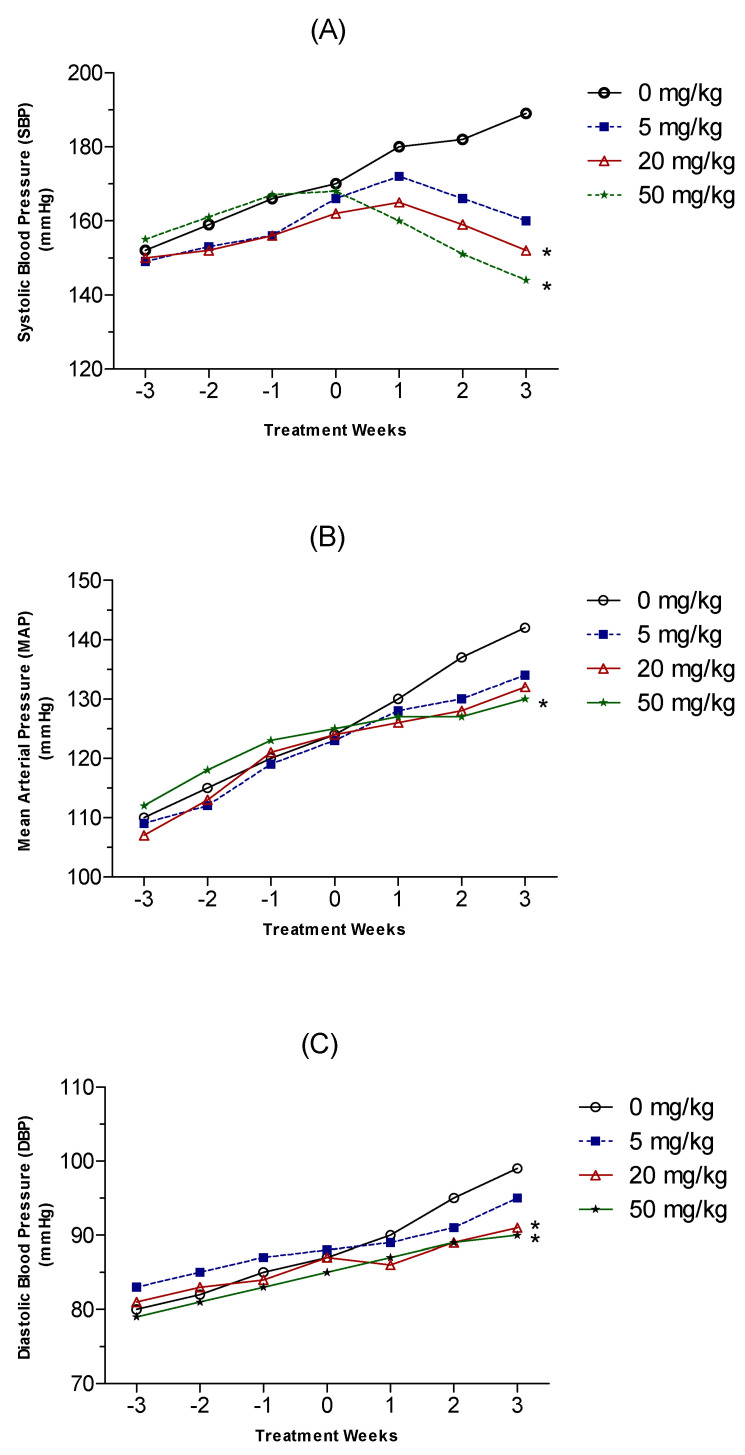
Average weekly systolic (**A**), mean arterial (**B**), and diastolic (**C**) blood pressure of control and HON-treated spontaneously hypertensive male rats (SHR) during the three-week study period. Data are presented as mean ± SD (*n* = 8). * indicates significant difference, lower or higher, from control.

**Figure 4 molecules-27-03396-f004:**
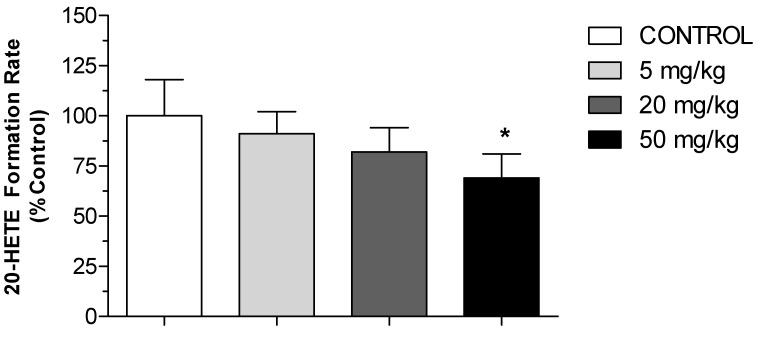
Effect of HON on the rate of 20-HETE formation in rat kidney microsomes, expressed as the percentage of the control group. Rat kidney microsomes obtained from SHR (*n* = 8/group) treated intraperitoneally daily for 3 weeks with HON at different doses. All groups were compared using one-way ANOVA followed by multiple comparisons. * Significant difference from control with *p* < 0.05.

**Figure 5 molecules-27-03396-f005:**
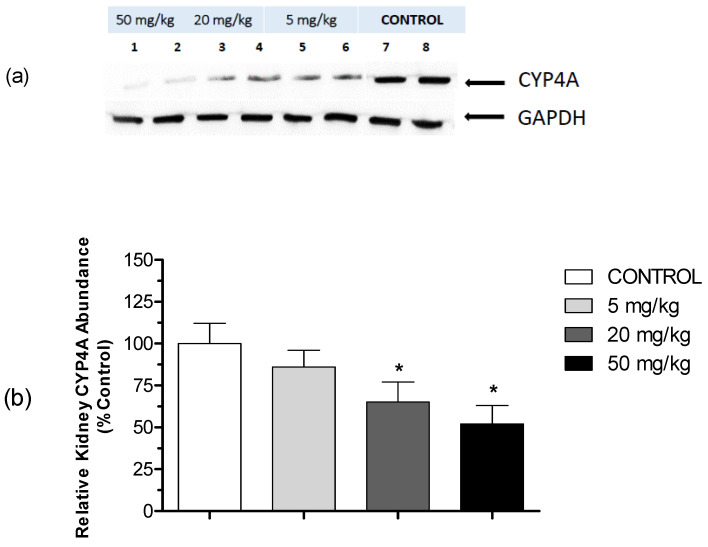
Effect of HON on kidney expression of CYP4A. The immunoreactive proteins were detected via enhanced chemiluminescence and x-ray film imaging, and the resultant signals were analyzed by densitometry. The intensity of the band was normalized to a loading control (GAPDH signal), then data were expressed as percentages of the control values. (**a**) The original specimen analyzed by densitometry (control, lanes 7–8; low-dose HON, lanes 5–6; medium-dose HON, lanes 3–4; and high-dose HON, lanes 1–2). (**b**) Renal CYP4A protein levels in rats (*n* = 8/group) treated intraperitoneally daily for 3 weeks with HON at different doses. Results are expressed as the percentages of control levels in untreated rats. All groups were compared using one-way ANOVA followed by multiple comparisons. * Significant difference from control with *p* < 0.05.

**Figure 6 molecules-27-03396-f006:**
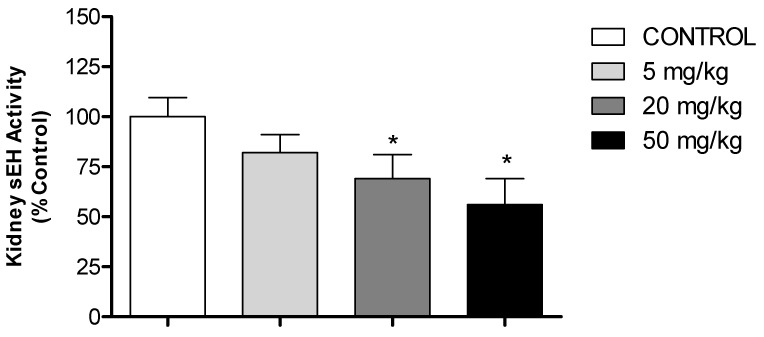
Effect of HON on the activity of kidney soluble epoxide hydrolase (sEH). Mean sEH activity from rat kidney cytosols (*n* = 8/group) treated intraperitoneally daily for 3 weeks with HON at different concentrations. All groups were compared using one-way ANOVA followed by multiple comparisons. * Significant difference from control with *p* < 0.05.

**Figure 7 molecules-27-03396-f007:**
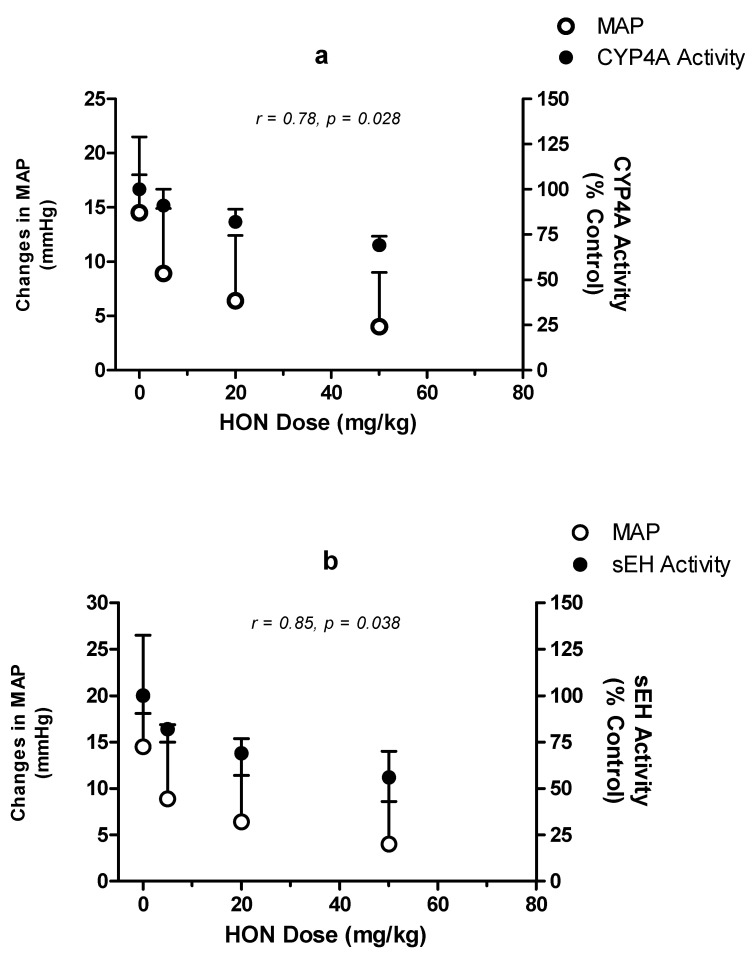
Correlation between changes in MAP, CYP4A activity, and sEH activity as a function of HON dose. Absolute changes in MAP (open circles) in correlation to CYP4A activity (closed circles, (**a**)) or sEH activity (closed circles, (**b**)) as a function of HON dose. MAP is presented as absolute change from baseline (week 0). Both CYP4A and sEH activities are presented as a percentages of control activity.

## Data Availability

Not applicable.

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
