# Peer review of "The Modulation of Arachidonic Acid Metabolism and Blood Pressure-Lowering Effect of Honokiol in Spontaneously Hypertensive Rats"

_molecules, 2022, doi:10.3390/molecules27113396_

Round 1

Reviewer 1 Report

The manuscript titled "Modulation of Arachidonic Acid Metabolism and Blood Pressure-Lowering Effect of Honokiol in Spontaneously Hypertensive Rats" by Elbarbry and Moshirian is a good work that provides valuable results of clinical relevance. The experiments are well planned and carried out, and the results are correctly presented. Only Fig. 3 should contain slightly larger, more readable graphs.

Author Response

REVIEWER #1

Comment: The manuscript titled "Modulation of Arachidonic Acid Metabolism and Blood Pressure-Lowering Effect of Honokiol in Spontaneously Hypertensive Rats" by Elbarbry and Moshirian is a good work that provides valuable results of clinical relevance. The experiments are well planned and carried out, and the results are correctly presented. Only Fig. 3 should contain slightly larger, more readable graphs

RESPONSE:

  • Thank you for the comment. The authors updated Figure 3 to make it more readable.

Reviewer 2 Report

Currently, the topic on the antihypertensive effect of honokiol (HON) is of great interest to the scientific community and falls in the scope of this journal. Honokiol has been specified as a novel alternative to treat various disorders such as liver cancer, neuroprotective, anti-spasmodic, antidepressant, antithrombotic, antimicrobial, analgesic properties and others. I found this manuscript clear, detailed, well organized, and well written. The abstract adequately describes the study, principal results and conclusions. Data are properly analyzed and interpreted to support the conclusions. Tables and pictures are satisfactory and interpreted correctly.

I think that the part in "Materials and methods" relating to the HPLC analysis (lines 340-343) should be clarified (specify better the eluents and the gradient used).

Therefore, I consider the manuscript suitable for publication.

There are only minor revisions to be made before publishing the manuscript:

Lines 6-7: specify the Department and address of the Institute

Line 7: “Moshiriann@ pacificu.edu” with the first lowercase letter

Line 10: “Hypertension” with the first lowercase letter

Lines 12 and 234:  replace “anti-hypertensive” with “antihypertensive”, to make the wording in the manuscript uniform

Line 87: write “Magnolia officinalis” and “Magnoliaceae” in italics

Line 277: correct “process”

Line 300: move the bracket from “(PEG400..” to “(v/v/v)”

Line 303: correct “0.2 µM membrane filter” with “0.2 µm membrane filter”.

Author Response

Currently, the topic on the antihypertensive effect of honokiol (HON) is of great interest to the scientific community and falls in the scope of this journal. Honokiol has been specified as a novel alternative to treat various disorders such as liver cancer, neuroprotective, anti-spasmodic, antidepressant, antithrombotic, antimicrobial, analgesic properties and others. I found this manuscript clear, detailed, well organized, and well written. The abstract adequately describes the study, principal results and conclusions. Data are properly analyzed and interpreted to support the conclusions. Tables and pictures are satisfactory and interpreted correctly.

RESPONSE: Thank you for the comment. We really appreciate your positive feedback

Comment:

I think that the part in "Materials and methods" relating to the HPLC analysis (lines 340-343) should be clarified (specify better the eluents and the gradient used).

RESPONSE:   The following paragraph is added to the HPLC section titled “Microsomal Arachidonic Acid (AA) Hydroxylation and Analysis of 20-HETE Formation “and is yellow highlighted in the trackchanges version.

AA and 20-HETE were resolved on an Agilent Eclipse Plus C18 column (4.6 x 250 mm; Agilent Technologies, Santa Clara, CA) with UV detection at 200 nm. Initial mobile phase composition was 45% acetonitrile in water with 0.1% acetic acid. Linear gradient (0.5%/min) was utilized over 30 min, followed by a sudden increase to 20%/min to 100% acetonitrile. Flow rate was maintained at 0.3 ml/min and column temperature at 40°C.

Comment:

Therefore, I consider the manuscript suitable for publication.

RESPONSE:   Thank you

There are only minor revisions to be made before publishing the manuscript:

Lines 6-7: specify the Department and address of the Institute

RESPONSE:   Done

Line 7: “Moshiriann@ pacificu.edu” with the first lowercase letter

RESPONSE:   Done

Line 10: “Hypertension” with the first lowercase letter

RESPONSE:   Done

Lines 12 and 234:  replace “anti-hypertensive” with “antihypertensive”, to make the wording in the manuscript uniform

RESPONSE:   Done

Line 87: write “Magnolia officinalis” and “Magnoliaceae” in italics

RESPONSE:   Done

Line 277: correct “process”

RESPONSE:   Done

Line 300: move the bracket from “(PEG400..” to “(v/v/v)”

RESPONSE:   Done

Line 303: correct “0.2 µM membrane filter” with “0.2 µm membrane filter”.

RESPONSE:   Done

Reviewer 3 Report

Elbarbry and Moshirian investigated the effects of Honokiol on arachidonic acid metabolizing enzymes and their association with the observed reduced blood pressure in spontaneously hypertensive rats. It was a good study but its presentation in the manuscript must be improved.

Major comments

  • The effects of honokiol in normal rats were not investigated. The data is important to see whether honokiol also has a similar in the normal rats which could be translated in human normal individuals.  
  • The results did not indicate whether there was any significant difference among the different doses of honokiol.
  • Figure 3B is redundant. The magnitude of increase/decrease can be mentioned in the text.
  • The description of the results in the Results section should not include introduction, methodology or discussion (lines 114, 138-144, 157-159, 178-181, 194-197)
  • Possible mechanism of action of honokiol in inhibiting the enzymes should be discussed.
  • Magnolia officinalis should be in italic and its authority name be provided. 
  • The r value and its statistical significance should be inserted in Figure 7.
  • How did you justify the use of the statistical parametric test? Did you run normality test?
  • What statistical test did you use for repeated measures of SBP, DBP and MAP?
  • Only approximately 30% of the references were from the last five years. Please improve it up to 60-70%.

Minor comments

  • Many grammatical, typo and spelling errors were noted (e.g. X-ray, not x-ray; St Louis, not ST Louis). Numbers should be spaced out from their units.
  • Many words were inappropriately capitalized.
  • Standardize the use of minutes or min.
  • All abbreviations should be defined at their first appearance. Subsequent appearance should only use the abbreviations, not in full definition.
  • Numbers < 10 should be spelled out.
  • Figure 1 should include sEH.
  • The source of chemicals should be provided, complete with the country of origin).
  • Line 346: Please use the symbol of mu, instead of u.

Author Response

REVEWER #3

Elbarbry and Moshirian investigated the effects of Honokiol on arachidonic acid metabolizing enzymes and their association with the observed reduced blood pressure in spontaneously hypertensive rats. It was a good study but its presentation in the manuscript must be improved.

Major comments

Comment:

The effects of honokiol in normal rats were not investigated. The data is important to see whether honokiol also has a similar in the normal rats which could be translated in human normal individuals. 

The results did not indicate whether there was any significant difference among the different doses of honokiol.

RESPONSE:   We agree with this reviewer if the main objective of the study was to investigate the antihypertensive effect of honokiol. However, we were more interested in examining the role of arachidonic acid metabolism in the blood pressure lowering effect. To acknowledge this limitation, we have added the lack of including normotensive rats to the limitation section. Throughout the study, we have discussed the significant differences among the honokiol doses in blood pressure, 20-HETE formation, and epoxide hydrolase activity.

Comment:

Figure 3B is redundant. The magnitude of increase/decrease can be mentioned in the text.

RESPONSE:   We respectfully disagree with the reviewer. We believe Figures 3B are important to  give the reader a better visual illustration of the impact of honokiol by comparing blood pressure values at baseline and end of study.  As mentioned in the text, SHR normally have progressive rise in their blood pressure. Therefore, even if honokiol dies not lower blood pressure compared to baseline, it was able to resist the progressive rise in blood pressure as indicated especially in the medium and high doses groups.

Comment:

The description of the results in the Results section should not include introduction, methodology or discussion (lines 114, 138-144, 157-159, 178-181, 194-197)

RESPONSE:   We have only indicated a very brief summary of the methodology to prepare the reader for the results section. We have removed unnecessary redundancy from the results sections.

Comment:

Possible mechanism of action of honokiol in inhibiting the enzymes should be discussed.

RESPONSE:   Although this was not the intent of the study, we have conducted a correlation study to show that the observed blood pressure lowering effect is at least in part due to inhibition of 20-HETE formation and/or sEH enzymes. It should be noted that that is no CYP4A or sEH knockout rat model which is the only way to be solid about this mechanism of action.  Currently, we also study the antioxidant effect of honkiol in the samples collected from thi study which can also add to the mechanism of the antihypertensive effect of honokiol.

Comment:

Magnolia officinalis should be in italic and its authority name be provided.

RESPONSE:   done

Comment:

The r value and its statistical significance should be inserted in Figure 7.

RESPONSE:   done

Comment:

Only approximately 30% of the references were from the last five years. Please improve it up to 60-70%.

RESPONSE:   We have included references that are relevant to the study. Please let us know which reference(s) that require immediate attention (i.e. deleted or added)

Minor comments

Many grammatical, typo and spelling errors were noted (e.g. X-ray, not x-ray; St Louis, not ST Louis). Numbers should be spaced out from their units.

Many words were inappropriately capitalized.

Standardize the use of minutes or min.

All abbreviations should be defined at their first appearance. Subsequent appearance should only use the abbreviations, not in full definition.

Numbers < 10 should be spelled out.

Figure 1 should include sEH.

The source of chemicals should be provided, complete with the country of origin).

Line 346: Please use the symbol of mu, instead of u.

RESPONSE:   We have addressed all of these minor concerns. Thanks for your great review

Round 2

Reviewer 3 Report

There were still many suggestions that were not addressed (authority name for the plant species, redundancy of Figure 4B, selection of references, result descriptions etc). The blood pressure should be analyzed using repeated measure ANOVA.
